# Effects of Roux-en-Y Gastric Bypass and Sleeve Gastrectomy on Non-Alcoholic Fatty Liver Disease: A 12-Month Follow-Up Study with Paired Liver Biopsies

**DOI:** 10.3390/jcm10173783

**Published:** 2021-08-24

**Authors:** Julie Steen Pedersen, Marte Opseth Rygg, Reza Rafiolsadat Serizawa, Viggo Bjerregaard Kristiansen, Nicolai J. Wewer Albrechtsen, Lise Lotte Gluud, Sten Madsbad, Flemming Bendtsen

**Affiliations:** 1Gastro Unit, Medical Division, Copenhagen University Hospital Hvidovre, Kettegaard Allé 30, 2650 Copenhagen, Denmark; marte.opseth.rygg@gmail.com (M.O.R.); lise.lotte.gluud.01@regionh.dk (L.L.G.); flemming.bendtsen@regionh.dk (F.B.); 2Department of Pathology, Copenhagen University Hospital Hvidovre, 2650 Copenhagen, Denmark; reza.serizawa@regionh.dk; 3Gastro Unit, Surgical Division, Copenhagen University Hospital Hvidovre, 2650 Copenhagen, Denmark; viggo.bjerregaard.kristiansen@regionh.dk; 4Department of Biomedical Sciences, Faculty of Health and Medical Sciences, University of Copenhagen, 2200 Copenhagen, Denmark; nicolai.albrechtsen@sund.ku.dk; 5Department of Clinical Biochemistry, Rigshospitalet, University of Copenhagen, 2200 Copenhagen, Denmark; 6NNF Center for Protein Research, Faculty of Health and Medical Sciences, University of Copenhagen, 2200 Copenhagen, Denmark; 7Department of Endocrinology, Copenhagen University Hospital Hvidovre, 2650 Copenhagen, Denmark; sten.Madsbad@regionh.dk

**Keywords:** non-alcoholic fatty liver disease, Roux-en-Y gastric bypass, sleeve gastrectomy, bariatric surgery, liver histology, non-alcoholic fatty liver disease activity score

## Abstract

Roux-en-Y gastric bypass (RYGB) improves, and can sometimes resolve, non-alcoholic fatty liver disease (NAFLD) and non-alcoholic steatohepatitis (NASH) but data based on histological assessment for the efficacy of sleeve gastrectomy (SG) in resolving NAFLD are sparse. Consequently, we aimed to compare the efficacy of RYGB vs. SG on NAFLD 12 months after surgery. In a prospective cohort study, 40 patients with obesity underwent bariatric surgery (16 RYGB and 24 SG). During surgery, a liver biopsy was taken and repeated 12 months later. NAFLD severity was evaluated using the NAFLD Activity Score (NAS) and Kleiner Fibrosis score. RYGB and SG patients were comparable at baseline. Mean (standard deviation, SD) NAS was 3.3 (0.9) in RYGB and 3.1 (1.4) in SG (*p* = 0.560) with similar degrees of steatosis, inflammation, and ballooning. Two RYGB patients, and six SG patients, had NASH (*p* = 0.439). Twelve months after surgery, NAS was significantly and comparably (*p* = 0.241) reduced in both RYGB (−3.00 (95% CI −3.79–−2.21), *p* < 0.001) and SG (−2.25 (95% CI −2.92–−1.59), *p* < 0.001) patients. RYGB patients had significantly more reduced (*p* = 0.007) liver steatosis (−0.91 (95% CI −1.47–−1.2) than SG patients (−0.33 (95% CI −0.54–−0.13) and greater improvement in the plasma lipid profile. Fibrosis declined non-significantly. NASH was resolved in seven of eight patients without a worsening of their fibrosis. RYGB and SG have similar beneficial effects on NAS and NASH without the worsening of fibrosis. RYGB is associated with a more pronounced reduction in liver steatosis.

## 1. Introduction

Non-alcoholic fatty liver disease (NAFLD) is recognized as a major and escalating cause of chronic liver disease in most countries globally [1,2,3,4], where an estimated 1-in-4 adults have NAFLD. NAFLD is closely associated with obesity and type-2 diabetes mellitus (T2DM).

The prevalence of NAFLD in patients with extreme obesity undergoing bariatric surgery has been reported to exceed 90%, with 25–35% also having non-alcoholic steatohepatitis (NASH) [5,6].

Roux-en-Y gastric bypass (RYGB) and sleeve gastrectomy (SG) are bariatric procedures that offer a very fast and sustained long-term weight loss [1], and RYGB has been shown to offer similar or slightly better improvement in cardiometabolic risk parameters, including T2DM remission [2,3,4,5,6]. The nearly comparable clinical outcomes in relation to weight loss and remission of T2DM are surprising given the marked anatomical differences between the two procedures [7].

The vast majority of data investigating the efficacy of bariatric surgery on NAFLD has been obtained from RYGB patients and, indeed, several studies have reported its beneficial effect on NAFLD and NASH [8,9,10,11], including fibrosis [8,10,12].

As SG has become the most frequently performed bariatric procedure [13], it is important to establish whether SG offers the same benefits as RYGB on NAFLD.

However, the importance of the anatomical differences between RYGB and SG in the remission of NAFLD has received little attention, with few studies having directly compared the effects of RYGB vs. SG using follow-up liver biopsies [14,15,16]. Of these studies, two were retrospective in design [14,15] and one was prospective and consisted of 30 individuals (10 RYGB and 20 SG) in an Indian cohort [16]. Consequently, more comparative data on the outcomes of RYGB and SG among patients with NAFLD and liver fibrosis, from studies with a prospective design and using paired liver biopsies, are needed. We investigated the effect of RYGB vs. SG using paired liver biopsies taken during the bariatric procedures and again 12 months later.

## 2. Materials and Methods

### 2.1. Study Subjects

Seventy severely obese study subjects scheduled to undergo either RYGB (*n* = 30) or SG (*n* = 40) at a Danish University Hospital were enrolled consecutively 1–2 weeks prior to surgery. Mode of surgery (RYGB or SG) was decided by the endocrinologists after individual assessment and discussion with each patient. During surgery, a baseline (wedged) liver biopsy was taken.

All study subjects fulfilled the Danish National Bariatric Guidelines which were 1) Body Mass Index (BMI) > 35 kg/m^2^ and at least one of the following comorbidities: dyslipidemia, T2DM), hypertension, sleep apnea, polycystic ovarian syndrome, arthrosis in the lower extremities; or 2) BMI > 50 kg/m^2^ (no obesity-related comorbidity required). In addition, and in concordance with the national guidelines, the study subjects (regardless of type of surgery) completed a mandatory dietician-monitored, diet-induced weight loss of 8% before surgery.

At the time of the study there was a tendency towards the recommendation of the RYGB procedure in patients with manifest T2DM, but the study subject was offered the opportunity to select the surgical procedure unless there were specific contraindications to one of the procedures.

All study subjects were screened for other etiologies of liver disease and had no history of alcohol overuse.

In the year after surgery, all study subjects were subjected to the same post-operative clinical follow-up program.

Patients with a NAFLD activity score (NAS) of 2 or higher and/or fibrosis grade 1 or higher at the baseline biopsy (*n* = 69) were screened for inclusion in the follow-up study, with repeated (percutaneous) liver biopsy 12 months after surgery. Of these, one study subject was lost to follow-up and seven were excluded, leaving 61 study subjects who were invited to enroll. Twelve declined; 49 provided informed consent, nine of whom were later excluded for various reasons (Figure 1), leaving a total of 40 completely paired liver biopsies. The data presented in this paper are based on those 40 paired biopsies at baseline and 12 months follow-up. At baseline, there were no clinical, histological, or biochemical differences between the 21 eligible study subjects with only a baseline biopsy and the 40 study subjects with paired biopsies in the present study (data not shown).

### 2.2. Study Investigations

The 40 participants were investigated at baseline (day of surgery) and after 12 months (median 12.0 months (IQR 11.25;12.0). Phenotypical and anthropometrical data were obtained. Blood samples were collected in the morning on the day of surgery (baseline) and in the morning prior to percutaneous liver biopsy, both after a minimum of 10 h of fasting.

### 2.3. Tissue Sampling

Baseline: All patients underwent laparoscopic surgery. Liver tissue was sampled as a wedged biopsy (approximately 1.5 g) from the margo inferior of the left side of the right liver lobe. Tissue was sampled immediately after induction of anesthesia and placement of trocars before the bariatric procedure. Tissue was carefully collected using an ultrasonic dissection device and the innermost part of the biopsy underwent histological scoring.

Twelve months after surgery: The repeat liver biopsies were sampled percutaneously (TruCut, needle diameter 1.2 mm) and ultrasonically guided under local anesthesia from the right liver lobe.

After sampling, tissue was promptly immersed in 2% paraformaldehyde for later histological preparation.

### 2.4. Liver Biopsies and Histological Examination

The biopsies were blindly and independently evaluated by three liver pathologists using the NAFLD Activity Score (NAS) [17]. NAS was calculated as the sum of steatosis (graded 0–3), inflammation (graded 0–3), and ballooning (graded 0–2) scores. A NAS of 5 or above was considered to indicate NASH. Liver fibrosis was staged from F0 to F4.

### 2.5. Ethics

The study protocols were approved by the Regional Scientific Ethics Committee (H-16030784 and H-16030782) in the Capital Region of Denmark. The study was conducted in accordance with the Declaration of Helsinki, and oral and written informed consents were obtained from all study participants.

### 2.6. Statistics and Calculations

Data were assessed for normal distribution and log-transformed when necessary. Comparisons between RYGB and SG at baseline and 12 months after surgery were made using the independent samples *t*-test or Fisher’s exact test. The paired *t*-test was used for comparing baseline and follow-up values within each group and the independent samples *t*-test was used to test significance for delta values in RYGB and SG. Simple linear regressions with model control were used to explore associations among the three predictors (% excess body weight loss (%EWL), delta Homeostatic Model Assessment of Insulin Resistance (HOMA-IR), and delta adiponectin to change in NAS (delta NAS)) in RYGB and SG.

Data are presented as means (standard deviation (SD)) or frequencies and mean difference (95% confidence intervals (CI)).

The percentage of excess body weight loss (%EWL) at 12 months was calculated as (((baseline BMI − follow-up BMI)/(baseline BMI-25)) × 100%).

The HOMA-IR was calculated as (fasting glucose (mg/dL) × insulin (mU/L)/405) [18].

All statistical analyses were performed using IBM SPSS Statistics 25 64-bit.

## 3. Results

### 3.1. Baseline 

All clinical, anthropometrical, and biochemical parameters were comparable between RYGB and SG individuals at baseline (*p*-values all above 0.05) and are presented in Table 1 (specific *p*-values not shown)

Metabolically, both groups were insulin resistant, with fasting glucose above the upper reference limit of 5.5 mmol/L (RYGB: 6.7 mmol/L (1.5), SG: 6.1 mmol/L (0.6), *p* = 0.175) and elevated HOMA-IR (RYGB: 4.4 (0.9), SG: 6.0 (3.1), *p* = 0.764). Both groups had comparable cholesterol and triglyceride levels that were within the reference range. Plasma low-density lipoprotein (LDL) cholesterol levels were highest in the SG group (2.7 mmol/L (1.0) mmol/L), but only borderline significant (*p* = 0.063) compared with LDL levels in the RYGB group (2.1 mmol/L (0.6)). The liver enzymes ALT and AST were within normal ranges in both groups.

Baseline liver histology in RYGB and SG is depicted in Figure 2A, Table 2 and Table 3. Based on liver histology, two subjects in the RYGB group, and six subjects in the SG group, fulfilled the histological criteria for NASH (*p* = 0.439). Overall, steatosis grades were higher in RYGB (0.9 (0.7)) than in SG (0.5 (0.7)) but this difference was not significant (*p* = 0.160). Mean NAS was 3.3 (0.9) in RYGB and 3.1 (1.4) in SG (*p* = 0.560). Grade of inflammation, ballooning, and fibrosis were similar in RYGB and SG.

### 3.2. Twelve Months after Surgery: Impact of RYGB and SG on Anthropometrical and Biochemical Profiles 

A pronounced reduction in BMI was observed 12 months after both RYGB and SG, with a marked improvement in metabolic, inflammatory, and adipokine parameters, in addition to blood pressure levels (Table 1). The %EWL was not significantly different between the two groups 12 months after surgery (RYGB: 70 (23) and SG: 62 (23), *p* = 0.092). Fasting plasma insulin, -C-peptide, and glucose levels, and consequently HOMA-IR, decreased significantly and similarly in both groups.

Among RYGB patients, there was an improvement in the lipoprotein profile, including a significant mean decrease in LDL of −0.6 mmol/L (95% CI −0.9–−0.3, *p* < 0.001), whereas LDL did not change between baseline and 12 months follow-up in the SG group (mean LDL decrease −0.01 mmol/L, 95% CI −0.4–0.4, *p* = 0.943). Consequently, LDL concentrations differed significantly between RYGB and SG 12 months after surgery (*p* < 0.001). Reductions in very-low density lipoproteins (VLDL) and triglyceride levels were also more pronounced in the RYGB group, with significant mean reductions in VLDL of −0.3 mmol/L (95% CI: −0.5–−0.2, *p* < 0.001) and triglycerides of −0.68 mmol/L (95% CI: −1.04–−0.32, *p* <0.001), whereas only negligible declines were observed in SG after 12 months (mean reduction in VLDL: −0.13 mmol/L (95% CI −0.29–0.04), *p* = 0.121; mean reduction in triglycerides: −0.29 mmol/L (95% CI −0.60–−0.03), *p* = 0.071). Yet, when comparing the delta changes, these were not significant (VLDL *p* = 0.074, triglycerides *p* = 0.056) and after 12 months the plasma levels of VLDL and triglycerides were not different between the RYGB and SG groups (VLDL *p* = 0.093, triglycerides *p* = 0.096).

Notably, ALT levels did not decrease after RYGB (mean reduction −1.0 U/L (95% CI −6.0–7.8, *p* = 0.791)) but decreased significantly after SG (mean reduction −11.0 U/L (95% CI −16.7–−5.1, *p* < 0.001)), which resulted in markedly higher ALT levels in RYGB 12 months after surgery (*p* = 0.006).

### 3.3. Effects of RYGB and SG on Liver Histology 

Liver histology improved after surgery, with a significant decrease in the NAS (RYGB: −3.00 (95% CI −3.79–−2.21), SG: −2.25 (95% CI −2.92–−1.59)) and all sub-scores (steatosis, inflammation, ballooning) in both RYGB and SG (Figure 2C,D). At 12 months follow-up, seven of the eight study subjects with NASH at baseline saw it resolved without a worsening of fibrosis, but one study subject (a SG patient) had progressed in fibrosis from stage 1 to stage 2. Although steatosis declined significantly after both RYGB (mean reduction −0.91 (95% CI −1.47–−1.2, *p* < 0.001)) and SG (mean reduction −0.33 (95% CI −0.54–−0.13, *p* = 0.003)), RYGB patients experienced a more pronounced reduction in liver steatosis, as delta steatosis was significantly greater in RYGB patients than in SG patients (*p* = 0.007). In all 16 RYGB patients, the steatosis grade was reduced to 0 (Figure 2B,C,E), whereas five SG patients had grade 1 steatosis at follow-up (Table 2 and Table 3 and Figure 2D). Consequently, the average steatosis grade was considerably lower among RYGB patients than SG patients 12 months after surgery (*p* = 0.022) (Figure 2B and Table 2). NAS, inflammation, and ballooning were significantly and comparably reduced in RYGB and SG, and scores were similar between the two groups 12 months after surgery.

Although fibrosis decreased slightly among both RYGB and SG patients, the changes were not statistically significant from baseline (RYGB: *p* = 0.104, SG: *p* = 0.096) (Figure 2C,D). Neither degree of weight loss (%EWL), decrease in HOMA-IR, or increase in adiponectin was associated with a reduction in NAS in either of the groups (data not shown).

## 4. Discussion

In the present study, we found resolution of NASH and overall significant improvement in NAS, steatosis, inflammation, and ballooning, all of which were independent of the type of surgery. Our results indicate that RYGB and SG are equally effective at reducing NAS when assessed 12 months after surgery.

Several prospective and retrospective studies, most of them evaluating the effect of RYGB, have demonstrated the substantial benefits of bariatric surgery in reducing NAS and fibrosis, and in reversing NASH [8,9,10,11,12,19,20,21]. In addition, a recent meta-analysis that included all bariatric procedures (RYGB, biliopancreatic diversion with duodenal switch, biliointestinal bypass, jejunoileal bypass, gastroplasty, adjustable gastric banding, and SG) showed a significant reduction in NAS in a majority of patients [22]. However, to the best of our knowledge, only three other studies [14,15,16] in addition to our own have compared RYGB and SG for their effects on NAFLD using paired liver histology.

Of the three aforementioned studies, those by von Schönfels et al. [14] and Froylich et al. [15] were retrospective and their follow-up liver biopsies were not sampled in a prospective, systematic protocol. In the von Schönfels et al. study, biopsies (34 SG and 19 RYGB) were taken on a clinical indication approximately six months after bariatric surgery during the steep weight loss phase [14]; in the Froylich et al. study (14 SG and nine RYGB) the indication for sampling was not reported and was performed 1.7 years (RYGB) and 1.2 years (SG) after bariatric surgery [15]. Overall, Froylich et al. found that RYGB improved NAS, steatosis, inflammation, ballooning, and fibrosis, whereas SG only had an effect on NAS and steatosis at follow-up [15]. However, in their study there were considerable baseline differences between the RYGB and SG groups. For example, the RYGB patients were, on average, 10 years older and significantly more obese. In addition, it appears that the NAS and fibrosis scores were notably higher among RYGB than SG patients at baseline; however, baseline comparisons between groups were not reported in the paper [15]. Von Schönfels et al. [14] reported a small, but similar and significant, reduction in NAS of approximately one point in both RYGB and SG patients, and concluded that the efficacy of the two surgical procedures is comparable. The third study, a prospective study by Praveen Raj et al. [16], included 30 paired liver biopsies (10 RYGB and 20 SG patients) taken during surgery and six months post-operatively in an Indian cohort. In contrast to Froylich et al., they found that SG patients had significantly higher NAS at baseline than RYGB patients. Additionally, SG reduced NAS, steatosis, inflammation, ballooning, and fibrosis significantly, whereas this was only the case for NAS and steatosis in their RYGB group. They concluded that both RYGB and SG improve NAFLD, as they found no difference in liver histology six months after surgery between the groups. However, comparisons of the delta changes between the two groups were not reported.

In summary, our prospectively obtained data from 40 paired biopsies confirm the main findings of earlier, similar studies [14,15,16], but our comparisons of the changes between baseline and 12 months follow-up in RYGB and SG patients provide more solid histological evidence that both bariatric procedures are, in fact, equally good at reducing NAS. A great strength of our results, and our overall conclusions, is that all clinical, biochemical, and histological characteristics were similar between RYGB and SG patients at baseline; as such, the improvement in liver histology in both groups was not confounded by any differences in baseline values between the groups.

As expected, weight and BMI decreased massively in both groups after surgery, but with no difference in either %EWL or the degree of metabolic improvement between the groups.

However, RYGB appeared to be superior to SG in terms of improving lipoproteins and reducing liver steatosis—a phenomenon that has not been reported previously. After RYGB, a more pronounced improvement in plasma LDL levels was observed compared with SG patients. RYGB was also associated with a significant decrease in triglyceride and VLDL levels, which only decreased negligibly in SG patients. That RYGB is associated with a more pronounced effect on lipoprotein profile one year after surgery has been established in a recent meta-analysis [6] but the mechanisms behind this observation are unknown, because the causality of the difference in effect on hepatic fat accumulation remains elusive. After both SG and, in particular, RYGB, there is an accelerated entry of nutrients into the small intestine, but postprandial absorption and metabolism of glucose, protein, and fat, in addition to the secretion of various gut hormones, differ after RYGB compared to SG [23], and several studies have found that after RYGB there is some malabsorption of fat; this could partly explain the lower LDL [24,25]. We have previously reported that the postprandial rise in plasma triglycerides seen in un-operated control subjects and after SG surgery is nearly absent in RYGB patients, and as such RYGB might also modulate the fat metabolism, and hence improve steatosis in the liver [26,27].

In addition, gut hormones play a critical role in regulating insulin secretion, fat metabolism, and lipid storage [28,29,30]. The secretion of glucagon-like peptide 1 (GLP-1) after RYBG surgery is approximately 10-fold higher than in un-operated control subjects and twice as high after SG [23]. Little is known of the effect of endogenously secreted GLP-1 on NAFLD, but GLP-1 receptor agonists inhibit de novo lipogenesis in the liver (i.e., from glucose) and may thereby reduce steatosis [31]. Although weight loss is the major regulator of liver fat, differences in hepatic fat metabolism, in combination with the variations in gut hormone secretion profiles, may have some important modulating effects on liver fat accumulation after surgery, be it RYGB or SG. Therefore, the gut hormone profiles following RYGB may be more advantageous for reducing liver steatosis.

Considering the similar histological improvement, it is puzzling that we observed a significantly decreased ALT in SG-operated individuals but unaltered ALT levels in RYGB-operated individuals. This finding has recently also been noted in a study that compared ALT levels 2 years after SG, RYGB, and One Anastomosis Gastric Bypass in a total of 4980 patients [32]. Here, SG was superior to both RYGB and OAGB in reducing ALT levels. We can only speculate as to why RYGB is possibly associated with a lesser decline in ALT levels when we observe the same degree of weight loss, glycemic control, and histological NAFLD improvement. Whether the malabsorption induced by RYGB, with its effects on physiological, hormonal, metabolic, and cellular responses, may counteract the beneficial effect of weight loss, could be a focus in new studies.

The primary strengths of the present study are its prospective design and analysis of both baseline and follow-up biopsies in a controlled setting. A limitation of the study is the small sample size and the lack of randomization; however, at baseline, there was little evidence of selection bias between the two patient groups, with the exception that the endocrinologists recommended RYGB in patients with manifest T2DM. A potential consequence was a higher number of study subjects with T2DM in the RYGB group. However, this was not significant and HOMA-IR in the two groups was quite similar,

The mandatory 8% weight loss before surgery may have impacted the NAS score (steatosis in particular) at baseline with potentially lower NAS than if they had not been subjected to the mandatory pre-operative weight loss. We can only speculate what the effect of RYGB vs. SG would have been, had study subjects had a higher NAS and hence more severe NASH at baseline. However, overall, we do not find the pre-operative weight loss to be a major limitation on the results and conclusions presented in this study because all study subjects underwent the same weight loss and the focus of this study was the impact of the type of surgery at 12 months follow-up.

Another limitation is the difference in the liver samples themselves; these samples were a wedged biopsy at baseline but a percutaneous biopsy at 12 months. Finally, the follow-up biopsy was optional, and some patients declined further participation; however, we found no significant differences between those patients who declined and those who agreed to a follow-up biopsy.

Finally, because fibrosis takes several years to fully improve, as shown by Lassailly et al. [12], from our 12 months follow-up data we cannot draw any conclusions on the efficacy of RYGB vs. SG on fibrosis resolution.

## 5. Conclusions

By evaluating paired liver biopsy material during initial surgery and again 12 months later, we demonstrated the equally beneficial effects of RYBG and SG in reducing NAS and reversing NASH. Although of less importance, RYGB appeared to better reduce liver steatosis and improve the plasma lipoprotein profile. Because SG is currently the most popular bariatric procedure, our data should help us better understand different surgical methods and their effects on NAFLD. SG appears to be an equally good alternative to RYGB in bariatric candidates with NAFLD, despite there currently being no NAFLD guidelines regarding the efficacy of bariatric surgery in treating NAFLD. Larger, longitudinal, randomized controlled trials are needed to evaluate the prolonged effect of RYGB and SG on NAFLD and NASH.

## Figures and Tables

**Figure 1 jcm-10-03783-f001:**
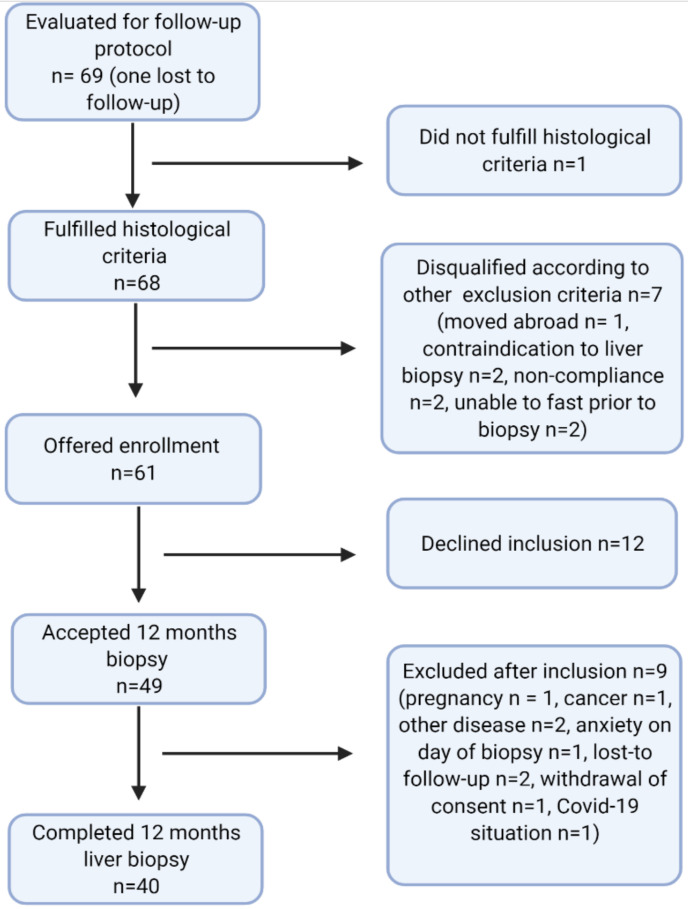
Flowchart of study subjects between baseline liver biopsy and 12 month follow-up liver biopsy.

**Figure 2 jcm-10-03783-f002:**
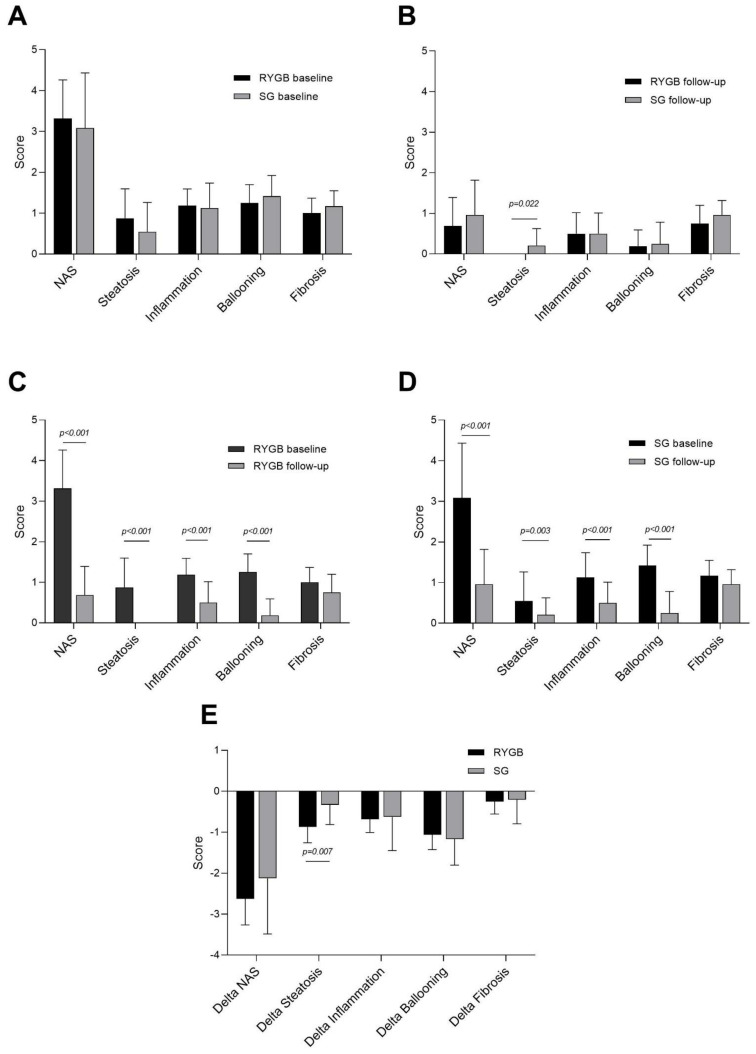
Liver histology in RYGB and SG at baseline and 12 months after surgery. NAFLD activity score (NAS) and the sub-scores for steatosis, inflammation, and ballooning, in addition to the fibrosis score in (**A**) Roux-en-Y gastric bypass (RYGB) vs. sleeve gastrectomy (SG) at baseline (day of surgery), (**B**) RYGB vs. SG 12 months after surgery, (**C**) in RYGB patients at baseline and 12 months after surgery, (**D**) in SG patients at baseline and 12 months after surgery, and (**E**) the delta changes (12 months after surgery—baseline) in RYGB vs. SG. Data are mean (SD), *n* = 40 (RYGB: *n* = 16, SG: *n* = 24).

**Table 1 jcm-10-03783-t001:** Clinical, anthropometrical, and biochemical characteristics at baseline and at 12 months follow-up.

	RYGB Baseline (*n* = 16)	RYGB Follow-Up (*n* = 16)	*p*-Value	SG Baseline (*n* = 24)	SG Follow-Up (*n* = 24)	*p*-Value	*p*-Value RYGB vs. SG Delta Changes	*p*-Value RYGB vs. SG Follow-Up
Age (years)	44 (2)	-	-	44 (9)	-	-	-	-
Female (*n*, %)	9 (56)	-	-	14 (58)	-	-	-	-
Diabetes (*n*, (%))	7 (44)	1 (6)		5 (21)	3 (12)			0.638
Weight (kg)	127 (24)	93 (19)	**<0.001**	123 (17)	95 (15)	**<0.001**	0.092	0.747
BMI (kg/m^2^)	43.0 (7.2)	31.4 (6.4)	**<0.001**	41.0 (4.5)	31.5 (4.0)	**<0.001**	0.363	0.966
% EWL	-	70 (23)	-	-	62 (23)	-	-	0.297
% total body weight loss	-	27 (7)	-	-	23 (9)	-	-	0.104
Waist-hip	0.94 (0.13)	0.88 (0.1	**<0.001**	0.91 (0.13)	0.86 (0.1)	**<0.001**	0.706	0.473
Systolic BP (mmHg)	129 (15)	119 (15)	**0.002**	128 (15)	122 (19)	0.148	0.288	0.499
Diastolic BP (mmHg)	83 (8)	78 (11)	**0.002**	83 (9)	78 (11)	**0.036**	0.200	0.162
Heart rate (BPM)	74 (16)	60 (11)	**<0.001**	74 (16)	62 (12)	**<0.001**	0.565	0.434
ALT (U/L)	33 (14)	32 (14)	0.791	32 (11)	21 (10)	**<0.001**	0.026	**0.006**
AST (U/L)	25 (9)	27 (7)	0.285	25 (8)	26 (14)	0.285	0.397	0.825
Fasting glucose (mmol/L)	6.7 (1.5)	5.5 (0.5)	**0.002**	6.1 (0.6)	5.3 (0.5)	**<0.001**	0.739	0.450
C-peptide (pmol/L)	1160 (204)	749 (253)	**<0.001**	1245 (450)	856 (356)	**<0.001**	0.142	0.387
Fasting insulin (pmol/L)	110 (24)	54 (21)	**<0.001**	144 (68)	79 (44)	**<0.001**	0.313	0.085
HOMA-IR	6.2 (0.9)	1.9 (0.8)	**0.035**	5.9 (2.7)	2.7 (1.8)	**<0.001**	0.281	0.127
LDL cholesterol (mmol/L)	2.1 (0.6)	1.5 (0.4)	**<0.001**	2.7 (1.0)	2.7 (1.0)	0.943	0.026	**<0.001**
HDL cholesterol (mmol/L)	1.21 (0.34)	1.4 (0.3)	**0.002**	1.22 (0.33)	1.5 (0.3)	**<0.001**	0.813	0.291
VLDL cholesterol mmol/L)	0.68 (0.35)	0.3 (0.1)	**<0.001**	0.63 (0.27)	0.5 (0.4)	0.121	0.074	0.093
Triglycerides mmol/L)	1.53 (0.77)	0.8 (0.2)	**<0.001**	1.37 (0.58)	1.1 (0.6)	0.071	0.056	0.096
HsCRP (mg/L)	7.4 (7.6)	1.5 (1.3)	**0.004**	3.9 (2.5)	1.3 (1.3)	**<0.001**	0.037	0.789
Adiponectin	6091 (1526)	12453 (7741)	**0.005**	5757 (2832)	10397 (4266)	**<0.001**	0.337	0.333
Leptin	44 (30)	22 (21)	**<0.001**	42 (26)	18 (15)	**<0.001**	0.720	0.507
Il-6	1.12 (0.67)	0.7 (0.3)	**0.011**	1.03 (0.7)	0.6 (0.4)	**0.007**	0.557	0.859
TNF-α	2.18 (0.85)	2.0 (0.6)	0.372	1.97 (0.52)	1.8 (0.5)	0.440	0.445	0.524

RYGB, Roux-en-Y gastric bypass; SG, sleeve gastrectomy; BMI, body mass index; % EWL, % excess body weight loss; mmHg, millimeter mercury; BP, blood pressure; BPM, beats per minute; ALT, alanine aminotransferase; AST, aspartate transaminase; HOMA-IR, Homeostatic Model Assessment for Insulin Resistance; LDL, low-density lipoprotein; HDL, high-density lipoprotein; VLDL, very-low density lipoprotein; HsCRP, high sensitive c-reactive protein; IL-6, interleukin 6; TNF-α, tumor necrosis factor alpha.Data are presented as mean (SD). *p*-values are paired samples *t*-test, Fisher’s exact test or independent samples *t*-test.

**Table 2 jcm-10-03783-t002:** Liver histology at baseline and at 12 months follow-up in RYGB and SG.

	RYGB Baseline (*n* = 16)	RYGB Follow-Up (*n* = 16)	*p*-Value	SG Baseline (*n* = 24)	SG Follow-Up (*n* = 24)	*p*-Value	*p*-Value RYGB vs. SG Delta Changes	*p*-Value RYGB vs. SG Follow-Up
Liver histology								
NAFLD activity score	3.3 (0.9)	0.7 (0.7)	**<0.001**	3.1 (1.4)	1.0 (0.9)	**<0.001**	0.241	0.302
Steatosis	0.9 (0.7)	0.0 (0.0)	**<0.001**	0.5 (0.7)	0.2 (0.4)	**0.003**	0.007	**0.022**
Inflammation	1.2 (0.4)	0.5 (0.5)	**<0.001**	1.1 (0.6)	0.5 (0.5)	**<0.001**	0.796	1.000
Ballooning	1.3 (0.4)	0.2 (0.4)	**<0.001**	1.4 (0.5)	0.3 (0.5)	**<0.001**	0.625	0.692
Fibrosis	1.0 (0.4)	0.8 (0.4)	0.104	1.2 (0.4)	1.0 (0.4)	0.096	0.826	0.131

RYGB, Roux-en-Y gastric bypass; SG, sleeve gastrectomy. Data are presented as mean (SD). *p*-values are paired samples *t*-test or independent samples *t*-test.

**Table 3 jcm-10-03783-t003:** Frequency table of NAFLD sub scores and fibrosis grade at baseline and 12 months follow-up in RYGB and SG patients.

	RYGB Baseline (*n* = 16)	RYGB Follow-Up (*n* = 16)	SG Baseline (*n* = 24)	SG Follow-Up (*n* = 24)
Steatosis grade (0/1/2/3)	5/8/3/0	16/0/0/0	14/7/3/0	19/5/0/0
Inflammation grade (0/1/2)	0/13/3	8/8/0	3/15/6	12/12/0
Ballooning grade (0/1/2)	0/12/4	13/3/0	0/14/10	19/4/1
Fibrosis grade (0/1/2/3/4)	1/14/1/0/0	4/12/0/0/0	0/20/4/0/0	2/21/1/0/0
NAS total sum (0/1/2/3/4/5/6/7/8)	0/0/3/7/4/2/0/0/0	7/7/2/0/0/0/0/0/0	0/2/8/6/2/6/0/0/0	8/10/5/1/0/0/0/0/0

RYGB, Roux-en-Y gastric bypass; SG, sleeve gastrectomy; NAS, NAFLD activity score.

## Data Availability

The data presented in this study are available on request from the corresponding author. The data are not publicly available due to restrictions set by the Danish Data Protection Agency.

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
