# Peer review of "Effects of Roux-en-Y Gastric Bypass and Sleeve Gastrectomy on Non-Alcoholic Fatty Liver Disease: A 12-Month Follow-Up Study with Paired Liver Biopsies"

_jcm, 2021, doi:10.3390/jcm10173783_

Round 1

Reviewer 1 Report

Thank you for the opportunity to review this paper. The authors investigated the change in liver histology in patients after bariatric surgery comparing sleeve gastrectomy vs. RYGB. Since this is only the 3rd study comparing SG vs. RYGB histologically, the paper is of interest and relevance. 

The paper is very well presented and well discussed in relation to the current evidence. 

I only have a few remarks that can be addressed in the discussion and by a change in the conclusion: 

  1. The authors point out several times that RYGB has a better effect on steatosis. I think this is an overstatement of the findings. This is a marginal difference and a NAS up to 2 is normal. Furthermore, the relevance of steatosis without inflammation and fibrosis is completely irrelevant as has been shown by many large trials. Therefore, I suggest that the authors tone that point down.
  2. Furthermore, while the authors performed a prospective study, the number of patients is not that much bigger than with the others. Furthermore, there is still a selection bias in the selection of the operation which is not clear, on which basis which procedure was recommended. This must be discussed. 
  3. In line with this points, there is a trend to baseline differences such as HOMA-IR and more patients with diabetes in the RYGB group. This all may have an impact on the results which should be discussed as a limitation. 
  4. The follow-up is too short for an effect on fibrosis. As the 5-year follow-up data from Lille (Lassailly et al. showed), the impact on firbosis needs several years to manifest. 
  5. While you focus on the difference in steatosis, how do you explain the stronger reduction of ALT after SG? What is the clinical relevance of the lower ALT after SG? One could argue that less ALT means less liver injury/hepatocyte death which would be beneficial. This finding of lower ALT has been shown in several studies as well. 
  6. The liver injury overall is very mild with less than 5 patients with real NASH (NAS of 5 or higher). This is also a limitation since the more advanced the liver injury, the stronger the impact or difference may be. This should be discussed.

Author Response

Reviewer 1

Thank you for the opportunity to review this paper. The authors investigated the change in liver histology in patients after bariatric surgery comparing sleeve gastrectomy vs. RYGB. Since this is only the 3rd study comparing SG vs. RYGB histologically, the paper is of interest and relevance. 

The paper is very well presented and well discussed in relation to the current evidence. 

We thank reviewer 1 for the time spent reviewing the paper, for valuable and insightful input and for the positive comments and contribution to improvement of the paper.

I only have a few remarks that can be addressed in the discussion and by a change in the conclusion: 

  1. The authors point out several times that RYGB has a better effect on steatosis. I think this is an overstatement of the findings. This is a marginal difference and a NAS up to 2 is normal. Furthermore, the relevance of steatosis without inflammation and fibrosis is completely irrelevant as has been shown by many large trials. Therefore, I suggest that the authors tone that point down.

Answer: We agree that inflammation and ballooning (and fibrosis) are the most important features of NASH. Accordingly, we have changed the wording in the conclusion to tone down the significance of steatosis.

Change in manus (conclusion): 'Although of less importance RYGB appeared to better reduce liver steatosis and improve the plasma lipoprotein profile'

  1. Furthermore, while the authors performed a prospective study, the number of patients is not that much bigger than with the others. Furthermore, there is still a selection bias in the selection of the operation which is not clear, on which basis which procedure was recommended. This must be discussed. 

Answer: Thank you for your comment. It is correct that the number of study participants were not much bigger in our study than in the studies that we compare our results with. A strength is that there were no significant baseline differences between the SG and RYGB groups at baseline.

One of the other reviewers also commented on how patients are selected for RYGB vs SG.

In Denmark the distribution between RYGB and SG surgery at the time of the study period was about 40%:60%. The decision about type of operation was based on information given to  the individual patient provided  by the endocrinologist (specialist bariatric endocrinologist) with input from the bariatric senior surgeon.The endocrinologist tends to recommend RYGB  in a patient with manifest T2DM, but the patient has the option to choose SG instead. Ultimately, it is the patient’s decision unless there are direct contraindications to one of the procedures.

Change in MS: We have added information regarding choice of surgery in the methods section and in the discussion/limitation pointed out a potential risk of selection bias (more patients with T2DM had RYGB).

Methods: 'At the time of the study there was a tendency towards the recommendation of the RYGB procedure in patients with manifest T2DM, but  the study subject was offered the opportunity to select the surgical procedure unless there were specific contraindications to one of the procedures.'

Discussion: 'A limitation of the study is the small sample size and the lack of randomization; however, at baseline there was little evidence of selection bias between the two patient groups except that the endocrinologists recommended RYGB in patients with manifest T2DM. A potential consequence hereof was a higher number of study subjects with T2DM in RYGB group.'

  1. In line with this points, there is a trend to baseline differences such as HOMA-IR and more patients with diabetes in the RYGB group. This all may have an impact on the results which should be discussed as a limitation. 

Answer: Thank you for your comment. In line with question/answer 2, it is correct that more study subjects had T2DM at baseline in the RYGB group than in SG group. As mentioned above this has now been added to the discussion/limitation. However, the distribution of T2DM patients between groups was  not significant (p=0.166). HOMA-IR at baseline was 6.2 in RYGB and 5.9 in SG at baseline (p=0.764).

  1. The follow-up is too short for an effect on fibrosis. As the 5-year follow-up data from Lille (Lassailly et al. showed), the impact on firbosis needs several years to manifest. 

Answer: Thank you for this notion. We fully agree, and have now commented on this in the revised manuscript.

Change in manus: 'Finally, as fibrosis takes several years to fully improve as shown by Lassailly et al. (34) we can not from our 12 months follow-up data draw any conclusions on the efficacy of RYGB vs SG on fibrosis resolution.'

  1. While you focus on the difference in steatosis, how do you explain the stronger reduction of ALT after SG? What is the clinical relevance of the lower ALT after SG? One could argue that less ALT means less liver injury/hepatocyte death which would be beneficial. This finding of lower ALT has been shown in several studies as well. 

Answer: That is a very interesting  question, to which we have no definite answer. We have now expanded the discussion regarding this finding.

Change in manus: 'Considering the similar histological improvement, it is puzzling, that we observe a significantly decreased ALT in SG operated individuals but unaltered ALT levels in RYGB operated individuals. This finding has recently also been noted in a study that compared ALT levels 2 years after SG, RYGB and One Anastomosis Gastric Bypass, respectively (32) . Here SG was superior to both RYGB and OAGB in reducing ALT levels. We can only speculate as to why RYGB is possibly associated with a lesser decline in ALT levels when we observe same degree of weight loss, glycemic control and histological NAFLD improvement. Whether the malabsorption induced by RYGB with its’ effects on physiological-, hormonal-, metabolic- and cellular responses may counteract the beneficial effect of weight loss could be the focus in new studies.'

  1. The liver injury overall is very mild with less than 5 patients with real NASH (NAS of 5 or higher). This is also a limitation since the more advanced the liver injury, the stronger the impact or difference may be. This should be discussed.

Answer: We agree about the observation, and this is now mentioned as a potential limitation to our findings and discussed.

Change in manus: 'The mandatory 8% weight loss before surgery could have impacted the NAS score (steatosis in particular) at baseline with potentially lower NAS than if they had not been subjected to the mandatory pre-operative weight loss. We can only speculate what the effect of RYGB vs. SG would have been, had study subjects had a higher NAS and hence more severe NASH at baseline. Yet, overall we do not find the pre-operative weight loss to be a major limitation to the results and conclusions presented in this study as all study subjects underwent the same weight loss and the focus of this study is the impact of type of surgery 12 months after surgery.'

Reviewer 2 Report

The manuscript is well-written and well-conducted, and the results are very pertinent.

I just have some comments to be addressed:

I suggest that the Authors may add how the patients have been evaluated before surgery and what criteria have been observed to prescribe surgery rather medical protocol and what kind of surgery (probably only Endocrinologists were not sufficient to decide).

Usually, before surgery a very low caloric diet (VLCD) were prescribed to reduce rapidly weight and liver volume. If it was used in Centre of the study, the Authors have to add in Methods. Furthermore, VLCD, inducing important weight loss, can modify liver steatosis stage and, thus, this would be discussed in Limitation of the study.

Did behavioral modification, dietary counseling, or physical training differ among groups of patients undergoing RYGB or SG? Was there any dietary and physical training that the patients received after surgery?

3.2 paragraph: the word “twelve”, related to months is lacking.

Figure 2: eliminate “2” from each figure.

I suggest that Authors may italicize or bold the p-values that are significant in the table to make it easier and more eye catching for the reader as there are lots of data in the tables.

In the tables, showing p values of the differences between RYGB and LG at baseline may be more useful rather p-value RYGB vs SG in FU. Indeed, the differences RYGB vs SG in FU do not add more information than delta changes. On the other hand, it is very important that the two population were comparable at baseline, as Authors stated also in the Discussion. 

Author Response

Reviewer 2

The manuscript is well-written and well-conducted, and the results are very pertinent.

We thank reviewer 2 for the positive and valuable comments and contribution to overall improvement of our manuscript.

I just have some comments to be addressed:

1.I suggest that the Authors may add how the patients have been evaluated before surgery and what criteria have been observed to prescribe surgery rather medical protocol and what kind of surgery (probably only Endocrinologists were not sufficient to decide)

Answer: Thank you for your comment. In Denmark the Endocrinologists decide the mode of surgery with input from the Bariatric surgeons only if there are surgical contraindications for either SG or RYGB. However, the patient him/herself has a substantial impact on the choice of surgery. The endocrinologist normally recommends RYGB  in a patient with manifest T2DM, but the patient has the option to prefer SG. Ultimately, it is the patient’s decision unless there are direct contraindications to one of the procedures.

In the methods section, we have added information on criteria for surgery in Denmark as well as provided more details on the process of selection of the patients to the two procedures, RYGB and SG. 

Change in manus: 'At the time of the study there was a tendency towards the recommendation of the RYGB procedure in patients with manifest T2DM, but  the study subject was offered the opportunity to select the surgical procedure unless there were specific contraindications to one of the procedures. All study subjects fullfilled the Danish National Bariatric Guidelines which were 1) BMI>35 kg/m2 and at least one of the following comorbidities: dyslipidemia, DM2, hypertension, sleep apnea, polycystic ovarian syndrome, arthosis in lower extremities or 2) BMI>50 kg/m2 (no obesity related comorbidity required). In addition, and in concordance with the national guidelines study subjects (regardless of type of surgery) completed a mandatory dietician-monitored, diet-induced weight loss of 8% before surgery. All study subjects were screened for other etiologies of liver disease and had no history of alcohol overuse. In the year after surgery all study subjects were subjected to the same post-operative clinical follow-up program.'

2.Usually, before surgery a very low caloric diet (VLCD) were prescribed to reduce rapidly weight and liver volume. If it was used in Centre of the study, the Authors have to add in Methods. Furthermore, VLCD, inducing important weight loss, can modify liver steatosis stage and, thus, this would be discussed in Limitation of the study.

Answer: Thank you for this important notion. Yes, patients were subjected to diet restriction prior to surgery as a part of the mandatory 8% weight loss that is required before both SG and RYGB surgery in Denmark (National bariatric guidelines). The 8% diet-induced weight loss is mentioned in the methods section and more details have now been provided. The weight loss is monitored by the endocrinologists and the dieticians in the department. It is not a strict VLCD but diet (rather than exercise) is central to the weight loss and the focus is to obtain a modest but steady weekly weight reduction. Weight loss of this size (8%) is typically achieved in 3-6 months in our department.

We do believe that this 8% weight loss has impacted the NAS (steatosis especially) at baseline and we are aware of this important limitation. But as all study subjects were subjected to the same degree of weight loss using the same diet principles prior to surgery and as the focus of this paper is the comparison of the magnitude of change in NAS and fibrosis score between SG and RYGB 12 months after surgery we have initially not mentioned it as a specific limitation to this study. However, we have now added this in the discussion/limitation (line 375-382)

Change in manus: 'The mandatory 8% weight loss before surgery could have impacted the NAS score (steatosis in particular) at baseline with potentially lower NAS than if they had not been subjected to the mandatory pre-operative weight loss. We can only speculate what the effect of RYGB vs. SG would have been, had study subjects had a higher NAS and hence more severe NASH at baseline. Yet, overall we do not find the pre-operative weight loss to be a major limitation to the results and conclusions presented in this study as all study subjects underwent the same weight loss and the focus of this study is the impact of type of surgery 12 months after.'

3.Did behavioral modification, dietary counseling, or physical training differ among groups of patients undergoing RYGB or SG? Was there any dietary and physical training that the patients received after surgery?

Answer: All study subjects were subjected to the same program of dietary counseling by expert bariatric dieticians in our Dept. of Endocrinology. There was no difference in the type or amount of counselling and all bariatric study subjects were subjected to the same rules and regulations. After surgery patients received the same routine medical and dietary follow-up and overall the dietary recommendations post-surgery are the same except that RYGB individuals are advised to be more aware of foods that can cause dumping syndrome.

In the methods section we have now added, that there was no difference in diet or degree of councelling during the 8% pre-operative weight loss.

Change in manus: 'In addition, and in concordance with the national guidelines study subjects (regardless of type of surgery) completed a mandatory dietician-monitored, diet-induced weight loss of 8% before surgery. All study subjects were screened for other etiologies of liver disease and had no history of alcohol overuse. In the year after surgery all study subjects were subjected to the same post-operative clinical follow-up program.'

4.3.2 paragraph: the word “twelve”, related to months is lacking.

Answer: Thank you. This has been added.

  1. Figure 2: eliminate “2” from each figure.

Thank you. This has been changed accordingly

6.I suggest that Authors may italicize or bold the p-values that are significant in the table to make it easier and more eye catching for the reader as there are lots of data in the tables.

Answer: We agree. Significant p-values have now been bolded.

7.In the tables, showing p values of the differences between RYGB and LG at baseline may be more useful rather p-value RYGB vs SG in FU. Indeed, the differences RYGB vs SG in FU do not add more information than delta changes. On the other hand, it is very important that the two population were comparable at baseline, as Authors stated also in the Discussion. 

Answer: Thank you for your comment. In table 1 we did not add information of the specific p-values (which were all above 0.05)  at baseline between SG and RYGB individuals as we thought this would add too much complexity to the table and the table would become too large. If a specific p-value had been below 0.05 we would have added a symbol, however all clinical, histological and biochemical data were comparable between the two groups at baseline.

Change in MS: We have now added (line 147) that the specific p-values do not appear in table 1 to clarify this point. 'All clinical, anthropometrical and biochemical parameters were comparable between RYGB and SG individuals at baseline (p-values all above 0.05)  and are presented in Table 1 (specific p-values not shown).'

Reviewer 3 Report

In the present prospective study, Pedersen and colleagues compared the efficacy of sleeve gastrectomy (SG) and roux-en-Y gastric bypass (RYGB) for NAFLD/NASH resolution 12 months after surgery. The authors observed similar efficacy for the two surgical approaches; in detatil, RYGB appeared more effective in improving liver steatosis and blood lipid profile, while SG appeared more effective on biochemical activity (despite no difference on liver inflammation was observed at biospy). Although the the limited number of patients included (n = 40), the study is valuable, well designed and the manuscript is well written. Below, some minor comments:

1) 2.2. Study partecipants. Second biopsy was performed after a mean of 12.1 months (range 11 - 18). Can the authors report the datum as median and IQR? it will be more informative on the dispersion of the data compared to the simple range. Furthemore, do the authors believe that the different time span could have affected the outcome? Was there any differences in the time bewteen baseline and last biopsy between patients that underwent SG and those that underwent RYGB?

2)  3.2. Months after surgery. At lines 185-188, the authors reported that ALT levels decreased significantly in patients that underwent SG while not in those that underwent RYGB. Despite mean baseline ALT values were below 40 U/L in both group, can the authors add a comment on this aspect in the discussion section?

3)  Discussion. The authors missed an important reference on this topic (Lassailly G, Gastroenterology 2020). Please add few lines of discussion in the manuscript.

Author Response

Reviewer 3

In the present prospective study, Pedersen and colleagues compared the efficacy of sleeve gastrectomy (SG) and roux-en-Y gastric bypass (RYGB) for NAFLD/NASH resolution 12 months after surgery. The authors observed similar efficacy for the two surgical approaches; in detatil, RYGB appeared more effective in improving liver steatosis and blood lipid profile, while SG appeared more effective on biochemical activity (despite no difference on liver inflammation was observed at biospy). Although the the limited number of patients included (n = 40), the study is valuable, well designed and the manuscript is well written. Below, some minor comments:

We thank reviewer 3 for the time spent reviewing our manuscript. Thank you for your positive feedback and valuable comments.

2.2 Study partecipants: Second biopsy was performed after a mean of 12.1 months (range 11 - 18). Can the authors report the datum as median and IQR? it will be more informative on the dispersion of the data compared to the simple range. Furthemore, do the authors believe that the different time span could have affected the outcome? Was there any differences in the time bewteen baseline and last biopsy between patients that underwent SG and those that underwent RYGB?

Answer: Thank you for your comment. We chose to give the range in order to be transparent about the fact, that a couple of study subjects due to COVID-19 situation had their 12 months biopsy postponed.

Change in MS: We have now  reported the median (12.0 months) and IQR (11.25;12.0).

2.3 Months after surgery. At lines 185-188, the authors reported that ALT levels decreased significantly in patients that underwent SG while not in those that underwent RYGB. Despite mean baseline ALT values were below 40 U/L in both group, can the authors add a comment on this aspect in the discussion section?

Answer: Thank you. This was commented by another reviewer as well and is interesting. We have added a short discussion of this observation in the discussion section

Change in manus: 'Considering the similar histological improvement, it is puzzling, that we observe a significantly decreased ALT in SG operated individuals but unaltered ALT levels in RYGB operated individuals. This finding has recently also been noted in a study that compared ALT levels 2 years after SG, RYGB and One Anastomosis Gastric Bypass, respectively (32) . Here SG was superior to both RYGB and OAGB in reducing ALT levels. We can only speculate as to why RYGB is possibly associated with a lesser decline in ALT levels when we observe same degree of weight loss, glycemic control and histological NAFLD improvement. Whether the malabsorption induced by RYGB with its’ effects on physiological-, hormonal-, metabolic- and cellular responses may counteract the beneficial effect of weight loss, could be focus in new studies.'

The authors missed an important reference on this topic (Lassailly G, Gastroenterology 2020). Please add few lines of discussion in the manuscript.

Answer and change in MS: thanks for pointing out this important reference. This reference has now been added to the manuscript and also commented  in the discussion.